# Serum and cerebrospinal fluid host proteins indicate stroke in children with tuberculous meningitis

**Charles M. Manyelo**[1], **Novel N. Chegou**[1], **James A. Seddon**[2,3], **Candice I. Snyders**[1], **Hygon Mutavhatsindi**[1¤], **Portia M. Manngo**[1], **Gerhard Walzl**[1], **Kim Stanley**[1], **Regan S. Solomons**[2]*

**1** Division of Molecular Biology and Human Genetics, DST/NRF Centre of Excellence for Biomedical Tuberculosis Research, South African Medical Research Council Centre for Tuberculosis Research, Faculty of Medicine and Health Sciences, Stellenbosch University, Cape Town, South Africa, **2** Department of Paediatrics and Child Health, Faculty of Medicine and Health Sciences, Stellenbosch University, Cape Town, South Africa, **3** Department of Infectious Diseases, Imperial College London, London, United Kingdom

¤ Current address: Wellcome Centre for Infectious Disease Research in Africa, Institute of Infectious Disease and Molecular Medicine, University of Cape Town, Observatory, South Africa

* regan@sun.ac.za

**Data Availability Statement:** All relevant data are within the paper and its Supporting Information files.

## Abstract

### Introduction

Stroke is a common complication in children with tuberculous meningitis (TBM). Host proteins may give us insight into the mechanisms of stroke in TBM and serve as biomarkers for detection of stroke, however, they have not been widely explored. In this study, we compared the concentrations of cerebrospinal fluid (CSF) and serum proteins between children who had TBM-related stroke and children with TBM without stroke.

### Methods

We collected CSF and serum from 47 children consecutively admitted to the Tygerberg Academic Hospital in Cape Town, South Africa between November 2016, and November 2017, on suspicion of having TBM. A multiplex platform was used to measure the concentrations of 69 host proteins in CSF and serum from all study participants.

### Results

After classification of study participants, 23 (48.9%) out of the 47 study participants were diagnosed with TBM, of which 14 (60.9%) demonstrated radiological arterial ischemic infarction. The levels of lipocalin-2, sRAGE, IP-10/ CXCL10, sVCAM-1, MMP-1, and PDGF-AA in CSF samples and the levels of D-dimer, ADAMTS13, SAA, ferritin, MCP-1/ CCL2, GDF-15 and IL-13 in serum samples were statistically different between children who had TBM-related stroke and children with TBM without stroke. After correcting for multiple testing, only the levels of sVCAM-1, MMP-1, sRAGE, and IP-10/ CXCL10 in CSF were statistically different between the two groups. CSF and serum protein biosignatures indicated stroke in children diagnosed with TBM with up to 100% sensitivity and 88.9% specificity.

**Funding:** This study was supported by the South African Government through the Technology Innovation Agency (TIA) (awarded to NC), the South African Research Chairs Initiative (SARChi) in TB Biomarkers (Grant number 86535) (awarded to GW), the International Collaborations in Infectious Disease Research (ICIDR): Biology and Biosignatures of Anti-TB Treatment Response (Grant number 5U01IA115619) (awarded to GW), the National Research Foundation of South Africa (Grant number 109437) (awarded to RS), the European Union through the European and Developing Countries Clinical Trials Partnership (EDCTP2) (Grant number TMA2018SF-2470-TBMBIOMARKERS) (awarded to NC) and the South African Medical Research Council through its Division of Research Capacity Development under the Internship Scholarship Programme (Grand number 57020) (awarded to CM), from funding received from the South African National Treasury. The funders had no role in study design, data collection and analysis, decision to publish, or preparation of the manuscript.

**Competing interests:** NC, CM, GW and RS are listed as inventors on an International Patent Application entitled "Cerebrospinal fluid and blood-based biomarkers for the diagnosis of tuberculosis meningitis" (PCT/IB2019/054259), filing date: 23 May 2019. NC and GW are listed as inventors on another patent application entitled "Method for diagnosing tuberculous meningitis" (PCT/IB2015/052751), filing date: 15 April 2015. These applications do not generate any royalties for the inventors. These does not alter our adherence to PLOS ONE policies on sharing data and materials.

## Conclusion

Serum and CSF proteins may serve as biomarkers for identifying individuals with stroke amongst children diagnosed with TBM at admission and may guide us to understand the biology of stroke in TBM. This was a pilot study, and thus further investigations in larger studies are needed.

## Introduction

Tuberculous meningitis (TBM) is the most common form of central nervous system (CNS) tuberculosis, mainly affecting younger children and immunocompromised individuals, including those living with human immunodeficiency virus (HIV) [1]. The true burden of TBM is unknown but it is estimated that globally at least 100,000 individuals develop the disease each year [2, 3]. Untreated TBM is invariably lethal. Even when treated, childhood TBM has very poor outcomes with up to 20% risk of death and above 50% risk of neurological sequelae among survivors [4]. Stroke, demonstrated by computed tomography (CT) and/or magnetic resonance imaging (MRI), is one of the main complications of TBM and is associated with poor clinical outcome [5].

The occurrence of stroke has been reported in up to 57% of TBM patients, with mortality about three times higher in those with stroke compared to those without [6]. Higher incidence is reported in younger children and/or those with advanced stage of TBM [7]. Early diagnosis and initiation of anti-tuberculous treatment for prevention of infarction is crucial for improved clinical outcome in TBM. However, due to late presentation with advanced stage TBM in many patients, neuroimaging reveals already established infarction at admission [8]. Therapeutics to prevent the development of new infarcts or the evolution of existing ones would likely have substantial clinical benefits. A recent small trial of aspirin demonstrated reduction of new infarcts and death in those with confirmed TBM [9]. To avoid unnecessary adverse events, it would be important to target therapies such as aspirin and other antiplatelet agents in those that would benefit most from them.

Several studies have demonstrated the roles of inflammatory mediators in the pathogenesis of TBM. Upregulation of inflammatory mediators including tumour necrosis factor (TNF)-α, interferon (IFN)-γ, interleukin (IL)-1β, IL-6, IL-8 and IL-10 in the cerebrospinal fluid (CSF) of patients with TBM have been described [10–13], when compared to symptomatic controls. Furthermore, certain serum and CSF cytokines have been shown to be associated with disease outcomes in TBM [14, 15]. High serum and CSF levels of IL-4 and IL-1β correlates with presence of infarcts on MRI brain [15]. In a recent study, elevated lumbar and ventricular CSF TNF-α, macrophage inflammatory protein 1α, IL-6, IL-8, as well as markers of brain injury were associated with infarcts in patients with TBM [8].

Inflammatory proteins may provide insight into infarction in TBM patients. Given that neuroimaging is not available in many low resource settings, where most patients develop TBM, a blood- or CSF-based test that could indicate stroke could allow targeted therapeutics. In addition, if a host protein biosignature could be developed that could predict future stroke, therapy could be targeted to those individuals. Finally, a better understanding of the biology of arterial ischemic stroke could contribute to the development of new therapeutic and preventive strategies [6]. In this analysis, we used data from previous studies [16, 17] to assess the difference in levels of CSF and serum proteins among children with suspected meningitis, who were finally diagnosed with a) TBM and stroke, b) TBM without stroke and c) children

without TBM, 'not-TBM'. We further assessed the ability of host proteins and combinations of proteins to indicate stroke among children diagnosed with TBM.

## Materials and methods

### Study setting

We used existing CSF and serum host protein concentration data from children diagnosed with TBM or "not-TBM", from previous studies [16, 17]. Briefly, in these studies we enrolled participants between November 2016 and November 2017 at Tygerberg Academic Hospital, Cape Town, South Africa [16, 17]. Children with suspected TBM are referred from primary care day hospitals, district and secondary level hospitals to our institution to establish the diagnosis of TBM and to treat the complications associated with advanced disease (stage 2 and 3 TBM, e.g. hydrocephalus). Our sample is a representative of typical patients from the study community. Children were included in the study if they (1) had signs and symptoms suggestive of meningitis and required routine diagnostic assessment including lumbar puncture for CSF investigations (2) were between the ages of 3 months and 13 years, and (3) parents or legal guardians were willing to give informed consent [16, 17]. In children older than 7 years, assent was obtained if they had a normal level of consciousness, i.e., a Glasgow Coma Score (GCS) of 15/15. Children aged 13 years and older were excluded from the study. Failure to obtain written consent also excluded children from the study. This study was approved by the Health Research Ethics Committee of the University of Stellenbosch (N16/11/142), Tygerberg Academic Hospital, and the Western Cape Provincial Government.

### Classification of study participants

The study participants were classified as TBM cases ('definite' TBM and 'probable' TBM) and 'not TBM' group, based on a published research case definition, which combines clinical, radiological and laboratory characteristics [18]. The 'not TBM' group included children with alternative diagnosis (other forms of meningitis and no-meningitis) [16, 17]. None of the children in the 'not-TBM' group was treated for TBM. The study participants diagnosed with TBM were classified as TBM-related stroke and TBM without stroke (no-stroke) based on neuroradiological evidence of arterial ischemic infarcts on CT and/or MRI at baseline. Radiological arterial ischemic infarction was defined as neuroimaging evidence of infarction, i.e. interruption of blood flow eventually resulting in focal encephalomalacia. Mostly small areas of arterial ischemic infarction in the territory of the middle cerebral artery perforators i.e basal ganglia and internal capsule, were observed. When CT was performed established arterial ischemic infarcts were considered, and when MRI was performed both established and evolving arterial ischemic infarction were considered.

### Sample collection

As previously described [16, 17], we collected an additional 1 ml of CSF into a sterile tube and 1ml of blood into a BD Vacutainer® serum tube, during the collection of CSF and serum samples for routine diagnostic purposes. Samples were transported to the immunology research laboratory for processing and storage within an average of 2 hours from collection. Blood samples were centrifuged at 1200 x g for 10 minutes and serum was harvested. CSF samples were centrifuged in a biosafety level 3 laboratory at 4000 x g for 15 minutes and supernatant was harvested. All samples were stored at -80°C until measurement of analytes.

## Immunoassays

Concentrations of 69 host proteins in serum and CSF samples from study participants with TBM (with or without infarction) and 'not TBM' were determined in our previous studies [16, 17]. We evaluated 69 host proteins including markers that were previously investigated as biomarkers for TBM [16, 17, 19] and adult pulmonary TB [20–22]. Briefly, the 69 host proteins were measured in CSF and serum samples using enzyme-linked immunosorbent assay (ELISA) and multiplex immunoassay (Luminex), as previously reported (S1 Table) [16, 17]. The levels of Cathelicidin LL-37 in serum and CSF samples were evaluated using an ELISA kit purchased from Elabscience Biotechnology Inc. (catalog #E-EL-H2438).

All Luminex experiments were performed on the Bio Plex platform (Bio Rad Laboratories, Hercules, USA) in an ISO15189 accredited laboratory using the reagent kits purchased from Merck Millipore (Billerica, MA, USA) and R&D Systems Inc. (Biotechne®, Minneapolis, USA) [16, 17]. Data acquisition and analysis of median fluorescent intensity were done using the Bio Plex Manager Version 6.1 software (Bio Rad Laboratories). The laboratory staff performing the Luminex experiments were blinded to the clinical classification of the study participants. The values of analytes in the quality control reagents evaluated with the samples were within their expected ranges.

## Statistical analysis

Data for this study were analysed using Statistica (TIBCO Software Inc., CA, USA) and Graphpad Prism version 8 (Graphpad Software Inc., CA, USA). Differential expression of host markers between the three groups were evaluated using one-way analysis of variance (ANOVA), with Fisher's Least Significant Difference (LSD) post hoc testing to determine the differences between TBM-related stroke and TBM without stroke. Games-Howell post hoc test was used for analysis of host markers in which Levene's test of homogeneity revealed unequal variance between the groups. P-values <0.05 were considered significant. Correction for multiple testing was done using Benjamini-Hochberg with a false discovery rate of 20%. Receiver operating characteristic (ROC) curve analysis was used to investigate the abilities of biomarkers to indicate stroke amongst children with TBM. Maximum values of Youden's index were used to select the optimal cut-off values yielding highest sensitivities and specificities for each marker [23]. The abilities of combinations of different biomarkers in indicating stroke amongst children with TBM were assessed using general discriminant analysis (GDA), followed by leave-one-out cross-validation.

## Results

### Patient characteristics

We included 47 children on suspicion of meningitis; 23 were finally diagnosed with TBM (3 definite TBM and 20 probable TBM) [16, 17]. The other 24 children were diagnosed as "not-TBM" and included 2 children with bacterial meningitis, 2 children with viral meningitis and children with no-meningitis as described in Table 1. The median age of all study participants was 22 months (interquartile range [IQR]: 10.5–57.0) and 16.2% (6/37) of children with available HIV results were positive. Evidence of Bacillus Calmette-Guérin (BCG) vaccination was documented in 33 (70.2%) children. Of the 23 study participants diagnosed with TBM, 14 (60.9%) had evidence of stroke (Fig 1). The median age for children with TBM and stroke was 23.5 months (IQR: 11.0–40.0) and for the no-stroke group was 15.0 months (IQR: 5.0–27.0).

### Differentially expressed baseline CSF proteins

Of the 69 host proteins investigated in CSF samples, the levels of lipocalin-2, soluble receptor for advanced glycation end products (sRAGE), and interferon-gamma inducible protein (IP)

**Table 1. Clinical and demographic characteristics of the study participants.**

| | **All** | **TBM (n = 23)** | | **Not-TBM[a]** |
| | | **Stroke** | **No stroke** | |
|---|---|---|---|---|
| Number of participants | 47 | 14 | 9 | 24 |
| Definite TBM, n (%) | 3 (6.4) | 2/14 (14.3) | 1/9 (11.1) | - |
| Median age, months (IQR) | 22.0 (10.5–57.0) | 23.5 (11.0–40.0) | 15.0 (5.0–27.0) | 30.0 (9.0–96.0) |
| Males, n (%) | 30 (63.8) | 8 (57.1) | 5 (55.6) | 17 (70.8) |
| HIV infection, n/no. tested | 6/37 | 0/14 | 0/8 | 6/15 |
| TB contact in history, n (%) | 14 (29.8) | 7/14 (50.0) | 3/9 (33.3) | 4 (16.7) |
| **Admission Characteristics** | | | | |
| Symptoms duration, days (median, IQR) | 7.0 (2.0–14.0) | 7.0 (2.0–28.0) | 7.0 (2.0–7.0) | 3.0 (1.0–14.0) |
| Advanced stage TBM (IIb and III) | 12 (52.2) | 9 (64.3) | 3 (33.3) | - |
| Fever, n (%) | 17 (36.2) | 6 (42.9) | 2 (22.2) | 9 (37.5) |
| Vomiting, n (%) | 12 (25.5) | 4 (28.6) | 4 (44.4) | 4 (16.7) |
| Weight loss, n (%) | 11 (23.4) | 6 (42.9) | 3 (33.3) | 2 (8.3) |
| Seizures, n (%) | 19 (40.4) | 5 (35.7) | 4 (44.4) | 10 (41.7) |
| Cough, n (%) | 16 (34.0) | 5 (35.7) | 1 (11.1) | 10 (41.7) |
| Altered consciousness, n (%) | 19 (40.4) | 9 (64.3) | 6 (66.7) | 4 (16.7) |
| Raised intracranial pressure | 10 (21.3) | 9 (64.3) | 1 (11.1) | 0 |
| Hemiplegia | 13 (27.7) | 9 (64.3) | 0 | 4 (16.7) |
| **Other radiological features** | | | | |
| Hydrocephalus | 20 (42.6) | 13 (92.9) | 6 (66.7) | 1 (4.2) |
| Tuberculoma | 4 (8.5) | 4 (28.6) | 0 | 0 |

[a]The 'not-TBM' group were children with alternative diagnosis including other meningitis: bacterial meningitis (n = 2) and viral meningitis (n = 2), and no-meningitis: asphyxia (n = 1), autoimmune encephalitis (n = 1), febrile seizure (n = 3), Guillain Barre (n = 1), HIV encephalopathy (n = 1), focal seizures (n = 1), leukemia (n = 1), miliary TB (with lymphocytic interstitial pneumonitis) (n = 1), developmental delay (n = 1), breakthrough seizure (n = 1), gastroenteritis (caused by shock) (n = 1), idiopathic intracranial hypertension (IIH) (n = 1), viral gastroenteritis (adenovirus and rotavirus) and encephalopathy (n = 1), stroke (n = 1), mitochondrial diagnosis (n = 1), viral pneumonia (this included also severe acute malnutrition (SAM) and nosocomial sepsis) (n = 1), febrile seizure and acute gastroenteritis (n = 1), and others (n = 1). The table was adapted and modified from Manyelo et al [16, 17]. IQR: interquartile range.

-10 (CXCL10) were significantly higher in children who had TBM-related stroke compared to TBM without stroke, while the levels of soluble vascular cell adhesion molecule (sVCAM)-1, metalloproteinase matrix (MMP)-1, and platelet derived growth factor (PDGF)-AA were increased in children with TBM without stroke compared to TBM with stroke. In addition, we observed trends ($0.05 < $p-value$ \leq 0.09$) in increased levels of granulocyte-macrophage colony-stimulating factor (GM-CSF), D-dimer and Brain-derived neurotrophic factor (BDNF), and lower levels of ferritin and apolipoprotein CIII were observed in children who had TBM-related stroke compared to TBM without stroke. After correction for multiple testing using Benjamini-Hochberg procedure, significant differences were only observed for the concentrations of sVCAM-1, sRAGE, MMP-1, and CXCL10 (Table 2, Fig 2). The levels of 35 CSF proteins were statistically different between children with TBM-related stroke and the 'not TBM' group (S2 Table).

Using ROC curve to assesses the abilities of individual CSF biomarkers to indicate stroke among children with TBM at baseline, we obtained the area under the ROC curve (AUC) above 0.70 for lipocalin-2, sP-selectin, glial cell-derived neurotrophic factor (GDNF), sVCAM-1, sRAGE, apolipoprotein CIII, MMP-1, MMP-7, d-dimer, myoglobin, BNDF, complement factor H, PDGF-AB/BB, CXCL10, MIP-1α, ADAMTS13, SAA, PEDF, A1AT, MIP-1β, sICAM-1, apolipoprotein AI, PDGF-AA, and GM-CSF (S2 Table). The AUCs for lipocalin-2,

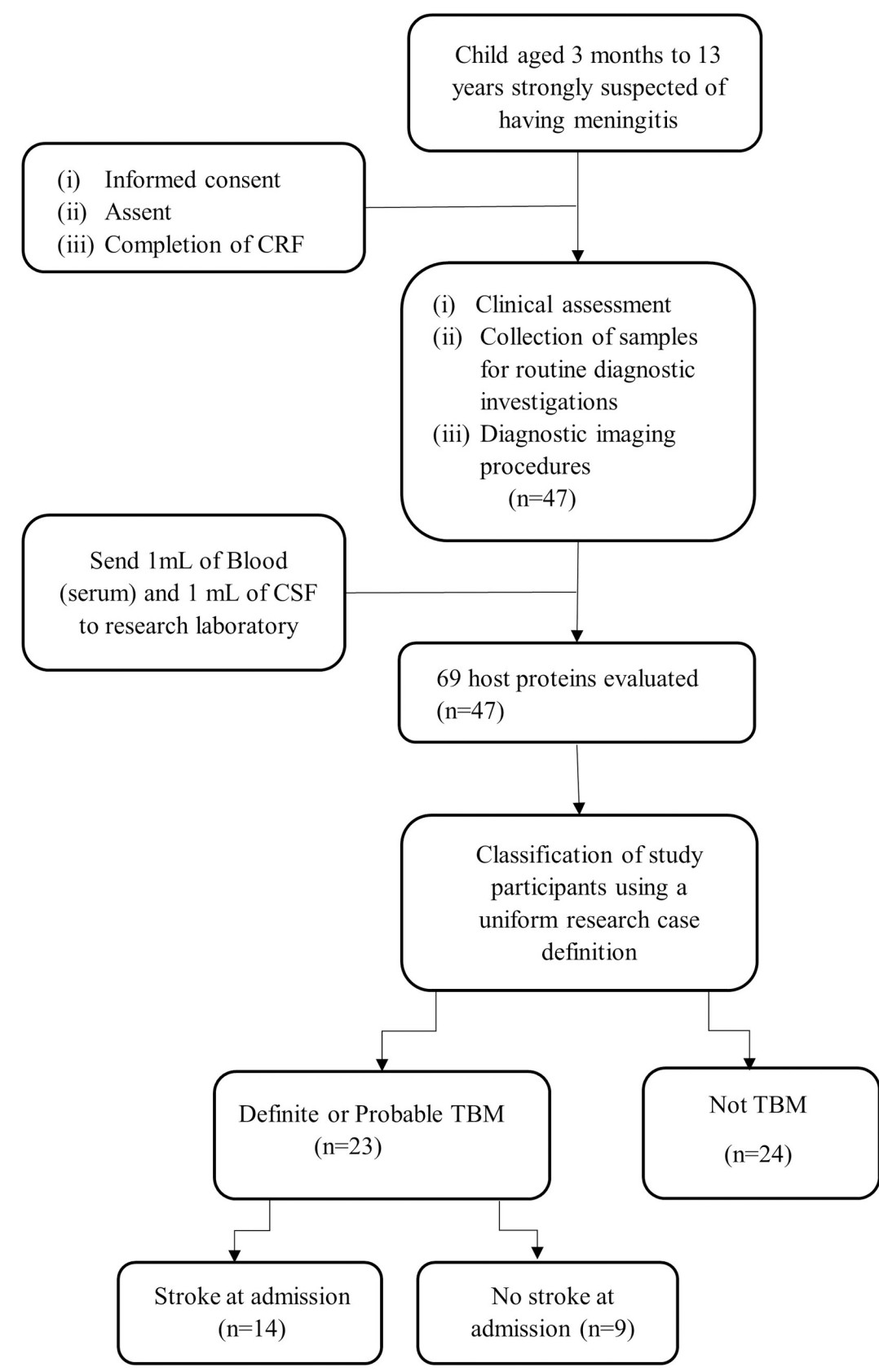

**Fig 1. Flow chart of the study design and classification of participants.** CRF: case report form; TBM: tuberculous meningitis; 'not TBM': children included on suspicion of meningitis and had an alternative diagnosis after investigations (other meningitis and no-meningitis). The 'not TBM' group included 2 children with bacterial meningitis, 2 with viral meningitis, and children with no-meningitis (Table 1). Children diagnosed with TBM were further classified as TBM with stroke and no stroke based on brain imaging findings at baseline.

sP-selectin, glial cell-derived neurotrophic factor (GDNF), sVCAM-1, sRAGE, MMP-7, D-dimer, MMP-1, Apolipoprotein CIII, myoglobin and BNDF were ≥0.75 in predicting stroke amongst children with TBM (Table 2, Fig 3).

## Utility of CSF protein combinations in indicating stroke amongst children with TBM

Data for all the CSF host proteins in children with TBM were analysed with GDA, regardless of HIV status, for investigation of combinations of proteins with optimal performance for indication of stroke. The most optimal model was obtained with a combination of four proteins, namely vascular endothelial growth factor (VEGF)-A, complement C5a, Complement factor 1, and BDNF. This four-protein biosignature indicated stroke among children with TBM with AUC of 0.98 (95% CI, 0.95–1.00), associated to sensitivity of 92.9% (95% CI, 66.1%-99.8%) (13/14) and specificity of 88.9% (95% CI, 51.8%-99.7%) (8/9). The positive predictive value (PPV) and negative predictive value (NPV) of the biosignature were 92.9% (95% CI,

**Table 2. Expression of CSF host protein biomarkers amongst study participants with TBM and stroke/no stroke at admission, and utility of individual CSF host protein biomarkers to indicate stroke in TBM patients.** The mean values shown (95% confidence intervals in brackets) are the least square (LS) means. Markers showing significant differences (p-value<0.05) or trends (0.05<p-value≤0.09) between the TBM patients with stroke and no stroke are shown. The differences in the concentrations of all other host markers are shown in S2 Table. #reported in ng/ml, all other markers are reported in pg/ml.

| Marker | TBM-stroke | TBM–no stroke | Not-TBM | P value[a] | S. by BH[b] | AUC (95% CI) | Cut-off value | Sensitivity% (95% CI) | Specificity % (95% CI) |
|---|---|---|---|---|---|---|---|---|---|
| sVCAM-1 | 83883.4 (50069.4–117697.3) | 161261.8 (119088.4–203435.2) | 64955.1 (39129.2–90780.9) | 0.0060 | Yes | 0.79 (0.60–0.98) | <120343.5 | 77.8 (45.3–96.1) | 71.4 (45.1–88.3) |
| MMP-1 | 493.5 (288.6–698.5) | 942.7 (687.0–1198.3) | 462.5 (306.0–619.0) | 0.0083 | Yes | 0.76 (0.53–0.99) | <480.355 | 77.8 (45.3–96.2) | 71.4 (45.4–88.3) |
| sRAGE | 14.8 (12.8–16.9) | 10.4 (7.8–12.9) | 14.8 (13.2–16.3) | 0.0086 | Yes | 0.78 (0.53–1.00) | >13.14 | 92.9 (68.5–99.6) | 66.7 (35.4–87.9) |
| CXCL10/IP-10 | 36270.7 (26910.6–45630.8) | 16554.0 (4879.9–28228.2) | 7338.6 (189.7–14487.5) | 0.0110 | Yes | 0.73 (0.51–0.95) | >7877.2 | 85.7 (60.1–97.5) | 66.7 (35.4–87.9) |
| PDGF-AA | 11.3 (7.3–15.3) | 18.4 (13.4–23.4) | 7.1 (4.0–10.1) | 0.0297 | No | 0.71 (0.46–0.95) | <18.29 | 55.6 (26.7–81.1) | 87.5 (64.0–97.8) |
| #Lipocalin-2/ NGAL | 118.7 (88.9–148.4) | 38.7 (1.6–75.8) | 11.1 (-11.6–33.9) | 0.0330 | No | 0.82 (0.63–1.00) | >45.88 | 78.6 (52.4–92.4) | 77.8 (45.3–96.1) |
| #Apolipoprotein CIII | 115.9 (30.0–201.8) | 248.3 (141.2–355.5) | 79.1 (12.0–146.1) | 0.0584 | No | 0.76 (0.54–0.98) | <85.14 | 88.9 (56.5–99.4) | 78.6 (52.4–92.4) |
| #D-dimer | 91101.8 (69953.6–112250.0) | 43555.6 (17179.2–69932.0) | 20614.5 (4462.4–36766.7) | 0.0645 | No | 0.76 (0.57–0.95) | >712.58 | 100.0 (78.5–100.0) | 55.6 (26.7–81.1) |
| BDNF | 0.9 (0.5–1.2) | 0.4 (-0.0–0.8) | 0.6 (0.3–0.9) | 0.0893 | No | 0.75 (0.50–1.00) | >0.175 | 92.9 (68.5–99.6) | 66.7 (35.4–87.9) |
| GM-CSF | 92.9 (75.11–110.6) | 66.0 (43.8–88.1) | 36.4 (22.9–50.0) | 0.0627 | No | 0.70 (0.43–0.98) | >71.08 | 85.7 (60.1–97.5) | 66.7 (35.4–87.9) |
| Ferritin | 6863.4 (3512.1–10214.6) | 11516.7 (7337.0–15696.4) | 3129.6 (570.1–5689.2) | 0.0870 | No | 0.66 (0.41–0.91) | <6166.45 | 66.7 (35.4–87.9) | 71.4 (45.4–88.3) |

[a]p-values shown are for post-hoc analysis between TBM with stroke compared to TBM without stroke.

[b]Benjamini-Hochberg procedure with false discovery rate (FDR) of 20%. Abbreviations: S. by BH: Significant by Benjamini-Hochberg

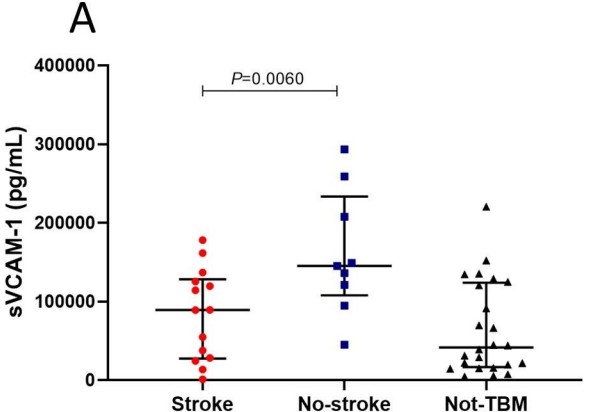

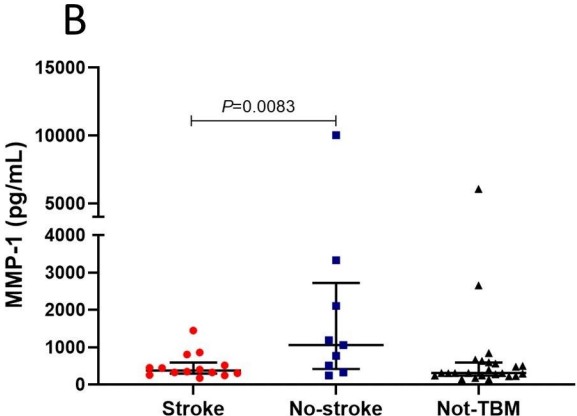

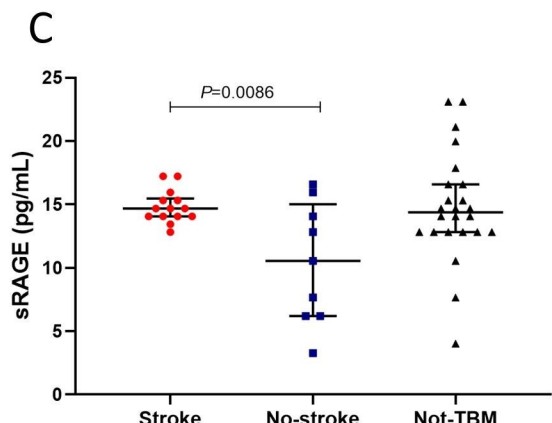

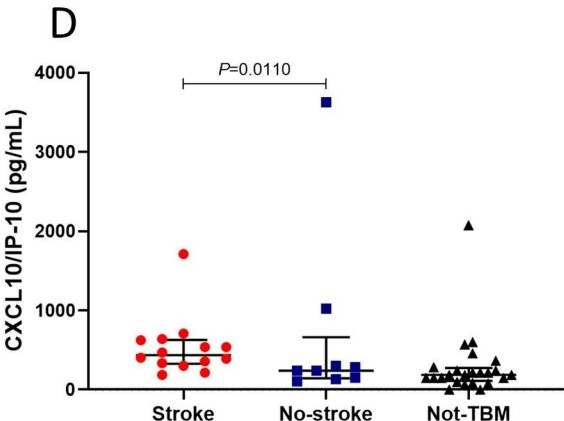

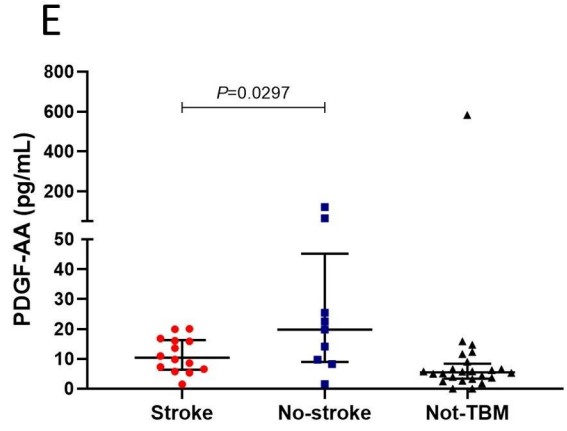

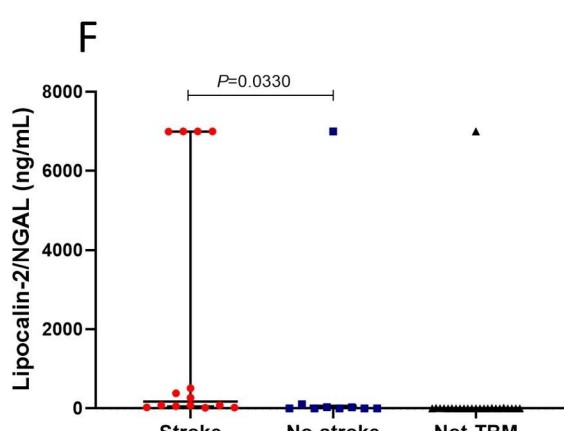

**Fig 2.** The concentrations of sVCAM-1 (A), MMP-1 (B), sRAGE (C), CXCL10/IP-10 (D), PDGF-AA (E), and lipocalin-2/NGAL (F) detected in cerebrospinal fluid samples from children who had TBM-related stroke and TBM without stroke (no-stroke). Horizontal bars depict median values and error bars are interquartile ranges. The p-values represent a comparison between TBM with stroke and TBM without stroke. The p-values shown were not corrected for multiple testing.

67.1%-98.8%) and 88.9% (54.4%-98.2%), respectively. Both the sensitivity and specificity of the four-protein biosignature remained the same after leave-one-out cross-validation (Fig 4).

## Differentially expressed baseline serum proteins

Of the 69 host proteins measured in serum samples, the levels of D-dimer, ADAMTS13, serum amyloid A (SAA), ferritin, monocyte chemoattractant protein (MCP-1)/CCL2 and growth differentiation factor (GDF)-15 were higher in children who had TBM-related stroke compared to the TBM without stroke group, whereas the concentrations of IL-13 were increased in

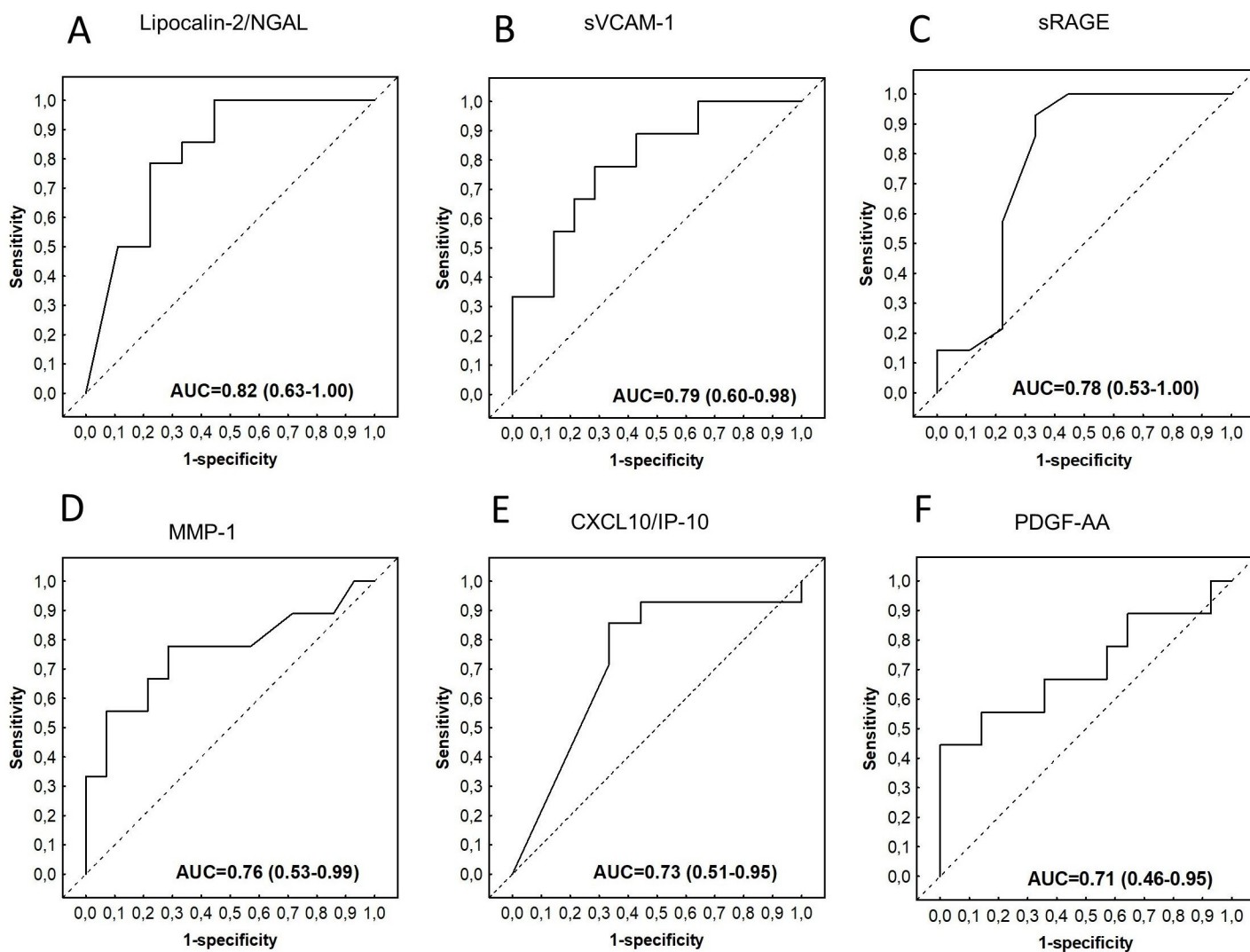

**Fig 3.** Receiver operator characteristic (ROC) curves showing the accuracies of baseline CSF lipocalin-2/NGAL (A), sVCAM-1 (B), sRAGE (C), MMP-1 (D), CXCL10/IP-10 (E), and PDGF-AA (F) in indicating stroke among children diagnosed with TBM. ROC curves for analytes with AUC≥0.70 are shown.

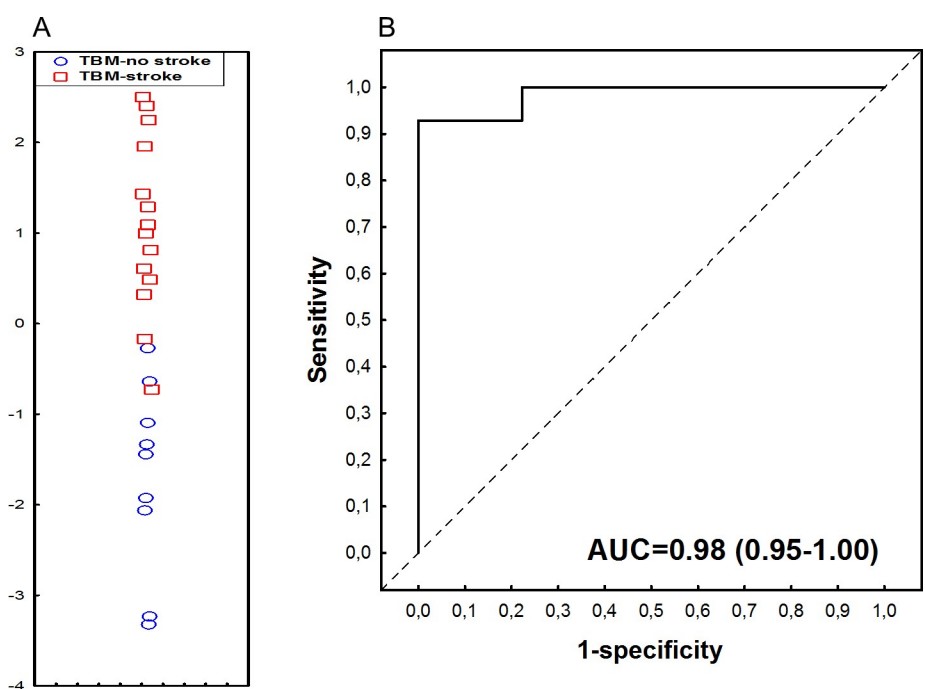

**Fig 4. Accuracy of the 4-marker CSF host protein biosignature (VEGF-A, complement component 5a, complement factor 1 and BDNF) in indicating stroke amongst children with TBM.** (A) Scatter plot depicting the separation of children as TBM with stroke/no-stroke using the 4-marker biosignature. (B) ROC curve depicting the performance of the 4-marker biosignature. Red squares: TBM-related stroke. Blue circles: TBM, no stroke.

children with TBM without-stroke compared to TBM with stroke. In addition, we observed trends (0.05<p-value≤0.09) in increased levels of GDNF, interleukin (IL)-7, MMP-9, lipocalin-2, IL-4, sP-selectin, and myoglobin, and lower levels of CC5a, in children with TBM-related stroke compared to TBM without stroke. After correction for multiple testing using Benjamini-Hochberg procedure, there was no statistical difference in the concentrations of the host proteins between children who had TBM-related stroke and TBM without stroke (Table 3, Fig 5). The levels of 14 serum proteins were significantly different between children with TBM-related stroke and 'not-TBM' group (S3 Table).

Assessment of the abilities of the individual biomarkers to indicate stroke using ROC curve analysis showed that 23 of the 69 proteins, namely GDF-15, D-dimer, ADAMTS13, MCP-'1, IL-4, CC4b, IL-7, ferritin, IL-10, SAA, I-309, CC5a, lipocalin-2, myoglobin, CC2, IL-13, RANTES, IL-1β, GDNF, serum amyloid P, MIP-1α, CC3, and PDGF-AB/BB indicated stroke in children with TBM with AUC ≥0.70 (S3 Table). Of note, the AUCs for GDF-15, D-dimer, ADAMTS13, CCL2/MCP-1, IL-4, CC4b, IL-7, ferritin, SAA, CCL1/I-309 and IL-10 were ≥0.75 in indicating stroke amongst children with TBM (Table 3, Fig 6).

## Utility of serum protein combinations in indicating stroke amongst children with TBM

The most accurate serum protein biosignature by GDA comprised IL-1β, IL-4 and alpha-1-Antitrypsin (A1AT), and indicated stroke amongst children with TBM with an AUC of 1.00 (95% CI, 1.00–1.00), associated to sensitivity of 100.0% (95% CI, 80.7%-100.0%) (14/14) and specificity of 88.9% (95% CI, 51.8%-99.7%) (8/9). The PPV and NPV of the biosignature were 93.3% (95% CI, 68.8%-98.9%) and 100.0% (95% CI, 59.8–100.0%), respectively. Following

**Table 3. Expression of serum host protein markers amongst study participants with TBM and stroke/no stroke at admission, and their accuracies in indicating stroke in TBM patients.** The mean values shown (95% confidence intervals in brackets) are the least square (LS) means. Markers showing significant difference (p-value<0.05) or showing trends (0.05<p-value≤0.09) between the TBM patients with stroke and no stroke are shown. The expressions of all other host markers are shown in S3 Table. #reported in ng/ml, all other markers are reported in pg/ml.

| Marker | TBM-stroke | TBM-no stroke | Not-TBM | P value[a] | S. by BH[b] | AUC (95% CI) | Cut-off value | Sensitivity % (95% CI) | Specificity % (95% CI) |
|---|---|---|---|---|---|---|---|---|---|
| #D-dimer | 4.1 (3.4–4.8) | 2.1 (1.2–2.9) | 3.6 (3.1–4.1) | 0.0187 | No | 0.83 (0.64–1.00) | >3.41004808 | 100.0 (78.5–100.0) | 66.7 (35.4–87.9) |
| IL-13 | 0.6 (-0.1–1.3) | 2.0 (1.1–2.9) | 1.3 (0.7–1.9) | 0.0194 | No | 0.73 (0.52–0.94) | <1.08834942 | 77.8 (52.4–92.4) | 78.6 (45.3–96.1) |
| #ADAMTS13 | 3.0 (2.7–3.3) | 2.1 (1.7–2.5) | 2.7 (2.5–2.9) | 0.0314 | No | 0.82 (0.63–1.00) | >2.90958695 | 85.7 (60.1–97.5) | 66.7 (35.4–87.9) |
| CCL2 | 2.5 (2.3–2.7) | 2.2 (2.0–2.4) | 2.8 (2.6–2.9) | 0.0322 | No | 0.82 (0.63–1.00) | >2.31141131 | 92.9 (68.5–99.6) | 66.7 (35.4–87.9) |
| #SAA | 4.9 (3.9–5.8) | 2.1 (0.9–3.3) | 3.9 (3.2–4.7) | 0.0325 | No | 0.75 (0.50–1.00) | >3.35686412 | 100.0 (78.5–100.0) | 66.7 (35.4–87.9) |
| Ferritin | 4.9 (3.9–5.8) | 2.2 (1.0–3.4) | 4.1 (3.3–4.8) | 0.0327 | No | 0.77 (0.53–1.00) | >4.22099783 | 92.9 (68.5–99.6) | 66.7 (35.4–87.9) |
| #GDF-15 | 0.4 (0.2–0.6) | 0.1 (-0.1–0.3) | 0.4 (0.3–0.6) | 0.0364 | No | 0.90 (0.78–1.00) | >0.296526673 | 78.6 (52.4–92.4) | 88.9 (56.5–99.4) |
| IL-4 | 1.8 (1.5–2.1) | 1.0 (0.6–1.4) | 2.1 (1.9–2.3) | 0.0673 | No | 0.81 (0.58–1.00) | >1.54370877 | 92.9 (68.5–99.6) | 77.8 (45.3–96.1) |
| IL-7 | 1.7 (1.5–1.8) | 1.4 (1.1–1.6) | 1.4 (1.2–1.5) | 0.0549 | No | 0.79 (0.59–1.00) | >1.46745807 | 85.7 (60.1–97.5) | 66.7 (35.4–87.9) |
| CC5a | 3.4 (3.3–3.5) | 3.6 (3.4–3.7) | 3.4 (3.3–3.5) | 0.0624 | No | 0.74 (0.51–0.96) | <3.41441601 | 88.9 (56.5–99.4) | 71.4 (45.4–88.3) |
| #Lipocalin-2/NGAL | 2.7 (2.2–3.3) | 1.3 (0.6–2.0) | 2.2 (1.8–2.7) | 0.0629 | No | 0.74 (0.47–1.00) | >2.56272926 | 71.4 (45.4–88.3) | 77.8 (45.3–96.1) |
| #Myoglobin | 1.2 (0.8–1.5) | 0.7 (0.3–1.1) | 1.2 (1.0–1.5) | 0.0856 | No | 0.74 (0.47–1.00) | >0.791936469 | 100.0 (78.5–100.0) | 66.7 (35.4–87.9) |
| GNDF | 2.2 (1.8–2.6) | 1.1 (0.6–1.6) | 1.8 (1.5–2.1) | 0.0518 | No | 0.72 (0.46–0.97) | >1.77128625 | 100.0 (78.5–100.0) | 55.6 (26.7–81.1) |
| #sP-selectin | 2.4 (1.9–2.9) | 1.2 (0.5–1.8) | 1.8 (1.4–2.2) | 0.0704 | No | 0.69 (0.41–0.96) | >1.00656614 | 100.0 (78.5–100.0) | 55.6 (26.7–81.1) |
| MMP-9 | 5.4 (4.4–6.5) | 2.6 (1.2–3.9) | 4.4 (3.6–5.2) | 0.0560 | No | 0.68 (0.39–0.98) | >5.13711673 | 85.7 (60.1–97.5) | 66.7 (35.4–87.9) |

[a]p-values shown are for post-hoc analysis between TBM with stroke compared to TBM without stroke.

[b]Benjamini-Hochberg procedure with false discovery rate (FDR) of 20%. Abbreviations: S. by BH: Significant by Benjamini-Hochberg.

leave-one-out cross-validation, the performance of the three-protein biosignature remained the same (Fig 7).

## Discussion

This study demonstrated that baseline CSF and serum host proteins are differentially expressed between children diagnosed with TBM, with stroke and no-stroke. Although only a few proteins showed statistical difference following correction for multiple testing, it is possible that other proteins which were statistically different prior to correction also have biological relevance as some have been associated with stroke in other studies. Individual CSF and serum host biomarkers, as well as combinations of proteins (biosignatures), demonstrated potential for indicating stroke amongst children diagnosed with TBM. The host biomarkers may be beneficial for early identification of stroke in TBM and timely clinical intervention to prevent poor clinical outcome or further deterioration. Given that neuroimaging is not available in

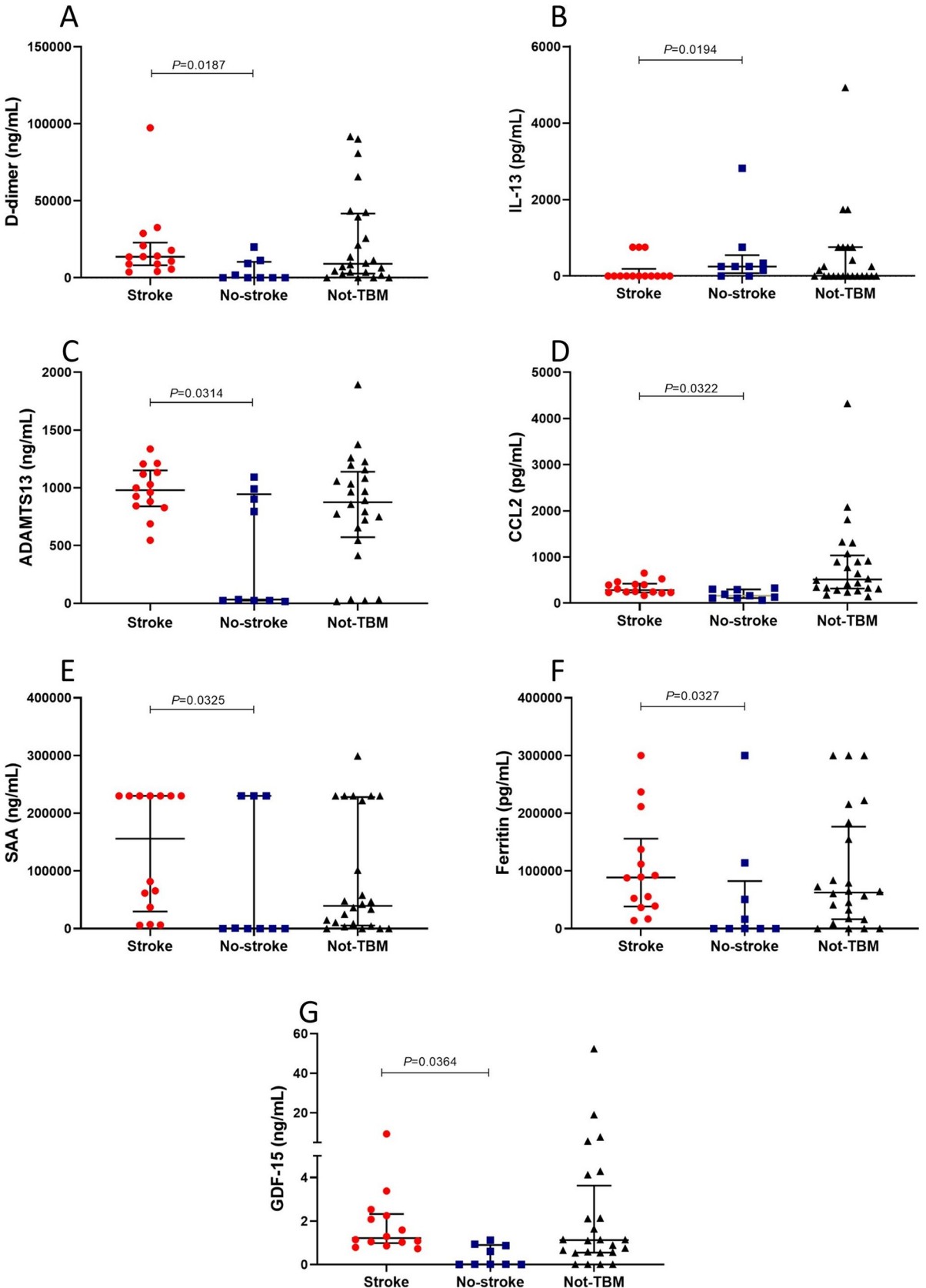

**Fig 5.** The concentrations of D-dimer (A), IL-13 (B), ADAMTS13 (C), CCL2/MCP-1 (D), SAA (E), Ferritin (F) and GDF-15 (G) detected in serum samples from children who had TBM-related stroke and TBM without stroke (no-stroke). Horizontal bars depict median values and error bars are interquartile ranges. The p-values represent a comparison between TBM with stroke and TBM without stroke. The p-values shown were not corrected for multiple testing.

many low resource settings, where most patients develop TBM, a blood- or CSF test based on host protein biomarkers will be beneficial. Such a test can be used to rapidly detect the presence of stroke in patients who are diagnosed with TBM, and thus initiate early appropriate therapy to prevent bad outcome.

The differentially expressed serum and CSF proteins described in this study may also help us to better understand the mechanisms of stroke in TBM for development of preventive and therapeutic strategies. Most of TBM pathology is attributed to the host inflammatory response [10–13]. A dysregulated inflammatory response in TBM contributes to formation of tuberculoma, obstruction of CSF flow and vascular complications including stroke [3]. Although studies have associated stroke with host proteins in TBM patients, their role is still unclear and requires further research. Our findings suggested the involvement of proteins associated with thrombus formation (such as D-dimer and ADAMTS13) [24, 25], proteins associated with

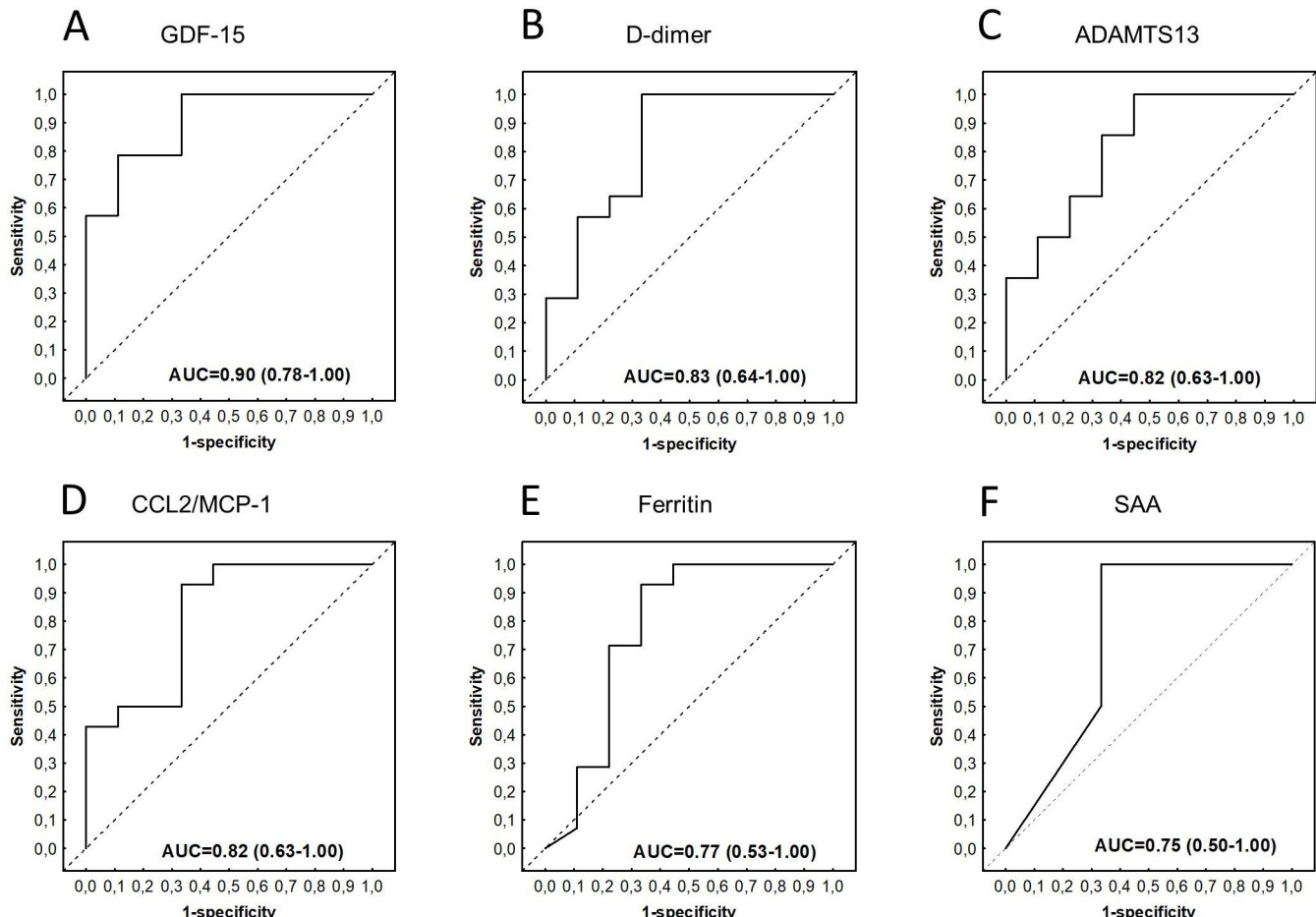

**Fig 6.** Receiver operator characteristic (ROC) curves showing the accuracies of baseline serum GDF-15 (A), D-dimer (B), ADAMTS13 (C), CCL2/MCP-1 (D), Ferritin (E), and SAA (F) in indicating stroke among children diagnosed with TBM. ROC curves for analytes with AUC≥0.75 are shown.

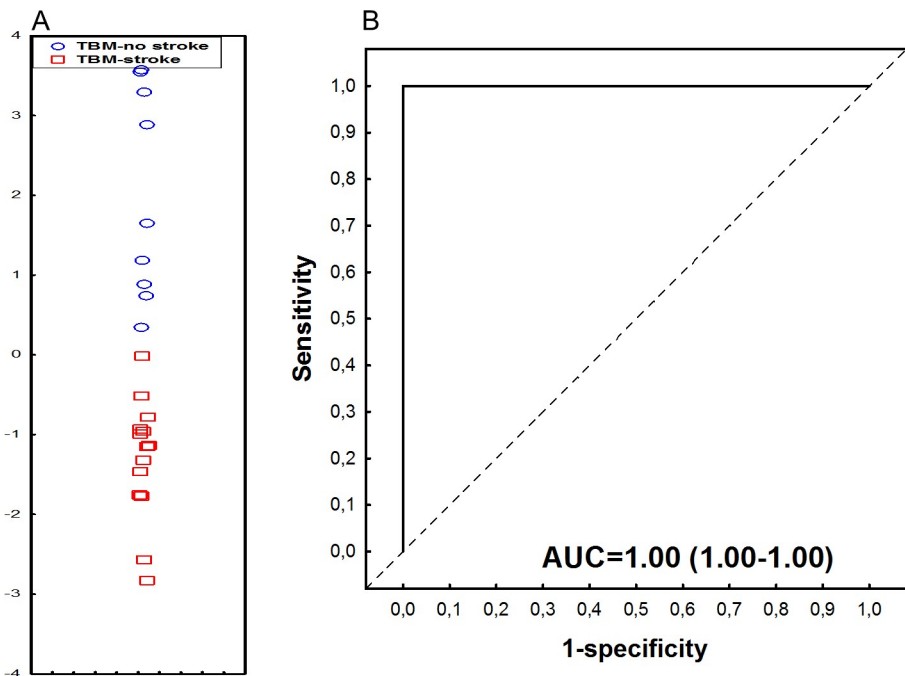

**Fig 7. Accuracy of the 3-marker serum host protein biosignature (IL-1β, IL-4, and Alpha-2-antitrypsin) in indicating stroke in children with TBM.** (A) Scatterplot depicting the separation of children as TBM with stroke or no stroke using the 3-marker biosignature. (B) ROC curve depicting the performance of the 3-marker biosignature. Red squares: TBM-related stroke. Blue circles: TBM, no stroke.

acute phase of ischemia (such as lipocalin-2, IP-10 and SAA) [26, 27], and angiogenic markers such as platelet derived growth factor (PDGF)-AA. However, the roles played by these proteins in the pathophysiology of stroke in TBM is still unknown. We observed lower levels of IL-13 in children with TBM and with stroke patients, which may suggest involvement of dysregulated inflammation. It was important to notice that following correction for multiple testing only the levels of sVCAM-1, sRAGE, MMP-1 and IP-10 in CSF demonstrated statistical significance. The upregulation of MMP-1 (collagenase-1) was previously described in ischemic brain after human stroke, and high levels of sVCAM-1 were reported in patients with brain infarctions [28, 29]. In contrast, we observed low levels of MMP-1 and sVCAM-1 in CSF patients with TBM stroke compared to the no-stroke group. In line with our findings, higher levels of IP-10 were previously detected within the ischemic region using human ischemic brain tissues [27].

A recent trial suggested that aspirin may reduce the incidence and promote resolution of TBM-associated stroke and inflammation, thus improving outcome [30]. Aspirin has anti-thrombotic and anti-inflammatory properties, and therefore could be used to target proteins involved in thrombus formation, such as D-dimer. In addition, aspirin may be helpful in reducing the inflammation associated with stroke by targeting pro-inflammatory proteins described in this study, such as lipocalin-2, IP-10 and SAA or by targeting and promoting anti-inflammatory proteins such as IL-13. Other anticoagulant (heparin) and antiplatelet agents could be targeted to prevent clot formation in TBM patients with increased clotting markers (D-dimer) at admission. ADAMTS13 is a well-described cleaving protease of von Willebrand factor (vWF), a key player in thrombus formation [24] and has been suggested a new therapeutic agent for promoting stroke recovery [31, 32]. We also suggest that

ADAMTS13 could be targeted to promote cleaving of vWF, thereby preventing thrombus formation and subsequent stroke. Angiogenesis is associated with tissue recovery after ischemic stroke [33], thus angiogenic factors including PDGF-AA could be targeted to promote resolution of TBM-related stroke.

Our study has some important limitations. The main concern is the small sample size, particularly of TBM patients with and without stroke. A further limitation is that we only evaluated the host proteins at one time point (baseline). Thus, changes in the expression of the proteins over the course of the disease or during treatment remains unknown. In addition, the association of host proteins with the severity, volume of infarcts or outcome remains to be investigated. It would also be necessary to assess the correlation of the host biomarkers described in this study with imaging (CT/MRI) findings and clinical characteristics such as age, gender, disease severity (TBM stage), and HIV status. We acknowledge that the lack of MRI imaging, which provides much more detailed neuroimaging than CT, is a limitation [34]. However, in TB endemic settings, the cost of MRI is often prohibitive. We definitely aim, with careful planning, to use MRI as the primary imaging modality in prospective studies. Previous studies have shown that TBM patients without infarcts at admission can show signs of infarction over the first weeks of treatment [8]. Thus, it would be good in the future study to assess evolution of infarcts over time and evaluate the abilities of biomarker signatures to predict the development of such infarcts, for preventive interventions. This could not be addressed in the current study as the patients were not followed up over time. Furthermore, it may be necessary to assess whether the concentrations of CSF host protein biomarkers are affected by the different head-down tilts done in ischemic stroke patients. Lastly, regarding the correction for multiple comparisons, it is acknowledged that an FDR of 20% may be large. The biosignatures reported in our study largely comprised biomarkers that were not significantly different between TBM-related stroke and no stroke, and this may be due to the smaller sample size. Thus the biomarkers comprising these biosignatures requires further assessment. However, this was a pilot study intended to explore host proteins, to identify candidates that may guide our understanding of the biology of stroke in TBM, and which could be useful as baseline biomarkers for detection of stroke at admission. We therefore identified candidate CSF and serum host protein biomarkers which could be further investigated in future larger studies.

## Conclusion

In summary, in addition to identifying candidate biomarkers and biosignatures which may be valuable as baseline detectors of stroke in patients diagnosed with TBM, and hence inform patient management practices, findings on the biomarkers evaluated in the current study may provide insight into biomarkers that are important in understanding the biology of stroke in TBM. Identification of patients with stroke at admission as shown in this study and/or early prediction of stroke if shown in future studies, may lead to timely appropriate treatment or the implementation of preventative or therapeutic strategies. Although findings of our study are potentially important, our study was preliminary, and the candidate biomarkers identified warrant further investigations in larger studies.

## Supporting information

**S1 File. The raw data for concentrations of host protein biomarkers measured in serum and cerebrospinal fluid samples from all the study participants using multiplex assay.** (XLSX)

**S1 Table. List of all host proteins evaluated in serum and cerebrospinal fluid using Luminex multiplex immunoassay and the suppliers of the reagent kits.**
(PDF)

**S2 Table. Expression of CSF host protein biomarkers amongst study participants with TBM and stroke/no stroke at admission, and their accuracies in indicating stroke in TBM patients.** The least square (LS) means (95% Confidence intervals) of all host markers and accuracies in indicating stroke amongst children with TBM are shown. #reported in ng/ml, all other markers are reported in pg/ml.
(PDF)

**S3 Table. Expression of serum host protein biomarkers amongst study participants with TBM and stroke/no stroke at admission, and their accuracies in indicating stroke in TBM patients.** The least square (LS) means (95% Confidence intervals) of all host markers and accuracies in indicating of stroke amongst children with TBM are shown. #reported in ng/ml, all other markers are reported in pg/ml.
(PDF)

## Acknowledgments

The authors are thankful to all the study participants and acknowledge the contribution made by the support staff.

## Author Contributions

**Conceptualization:** Novel N. Chegou, James A. Seddon, Gerhard Walzl, Regan S. Solomons.

**Data curation:** Charles M. Manyelo, Novel N. Chegou, Candice I. Snyders, Hygon Mutavhatsindi, Portia M. Manngo, Regan S. Solomons.

**Funding acquisition:** Novel N. Chegou, Gerhard Walzl, Regan S. Solomons.

**Investigation:** Charles M. Manyelo, Novel N. Chegou, Candice I. Snyders, Regan S. Solomons.

**Methodology:** Charles M. Manyelo, Novel N. Chegou, Candice I. Snyders, Gerhard Walzl, Regan S. Solomons.

**Project administration:** Novel N. Chegou, Kim Stanley, Regan S. Solomons.

**Resources:** Novel N. Chegou, Gerhard Walzl.

**Software:** Kim Stanley.

**Supervision:** Novel N. Chegou, Gerhard Walzl, Regan S. Solomons.

**Writing – original draft:** Charles M. Manyelo.

**Writing – review & editing:** Charles M. Manyelo, Novel N. Chegou, James A. Seddon, Candice I. Snyders, Hygon Mutavhatsindi, Portia M. Manngo, Gerhard Walzl, Kim Stanley, Regan S. Solomons.

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
