## [Decision Letter · Decision Letter 0]

12 Mar 2021

PONE-D-21-04815

Serum and cerebrospinal fluid host proteins predict stroke in children with tuberculous meningitis

PLOS ONE

Dear Dr. Solomons,

Thank you for submitting your manuscript to PLOS ONE. After careful consideration, we feel that it has merit but does not fully meet PLOS ONE’s publication criteria as it currently stands. Therefore, we invite you to submit a revised version of the manuscript that addresses the points raised by the reviewers, during the review process.

We look forward to receiving your revised manuscript.

Kind regards,

Katalin Andrea Wilkinson, PhD

Academic Editor

PLOS ONE

Journal Requirements:

2. In your Methods section, please provide additional information about the participant recruitment method and the demographic details of your participants. Please ensure you have provided sufficient details to replicate the analyses such as:

a) a description of any inclusion/exclusion criteria that were applied to participant selection,

b) a statement as to whether your sample can be considered representative of a larger population, and

c) a brief description of how participants were recruited in the original study.

4.We note that you have a patent relating to material pertinent to this article. Please provide an amended statement of Competing Interests to declare this patent (with details including name and number), along with any other relevant declarations relating to employment, consultancy, patents, products in development or modified products etc. Please confirm that this does not alter your adherence to all PLOS ONE policies on sharing data and materials, as detailed online in our guide for authors http://journals.plos.org/plosone/s/competing-interests by including the following statement: "This does not alter our adherence to  PLOS ONE policies on sharing data and materials.” If there are restrictions on sharing of data and/or materials, please state these. Please note that we cannot proceed with consideration of your article until this information has been declared.

5.We noticed you have some minor occurrence of overlapping text with the following previous publication(s), which needs to be addressed:

https://scholar.sun.ac.za/handle/10019.1/106647

In your revision ensure you cite all your sources (including your own works), and quote or rephrase any duplicated text outside the methods section. Further consideration is dependent on these concerns being addressed.

Reviewers' comments:

Reviewer's Responses to Questions

**Comments to the Author**

1. Is the manuscript technically sound, and do the data support the conclusions?

Reviewer #1: Partly

Reviewer #2: Partly

2. Has the statistical analysis been performed appropriately and rigorously? 

Reviewer #1: Yes

Reviewer #2: Yes

3. Have the authors made all data underlying the findings in their manuscript fully available?

Reviewer #1: Yes

Reviewer #2: No

4. Is the manuscript presented in an intelligible fashion and written in standard English?

Reviewer #1: Yes

Reviewer #2: Yes

5. Review Comments to the Author

Reviewer #1: Many thanks for asking me to review this paper. The manuscript describes analysis undertaken following your larger studies which have investigated a wider spectrum of protein changes in patients with TBM and those without. I agree with the authors that analysis of proteins related to stroke, an often disabling or life threatening sequelae in TBM, warranted further analysis. Knowledge gained here will contribute to our understanding of the mechanisms which underpin stroke pathogenesis within TBM. However, I have a few comments to be addressed:

1. Major: One of the conclusions of the proteins identified have the potential to contribute to management by 'predicting' stroke and therefore identifying those who may benefit from preventative therapy such as anti-platelet therapy. Given the design of this study where proteins were identified in blood and CSF samples of patients at baseline who either had or had not already developed stroke (ie imaging was performed at the same timepoint as blood/CSF samples taken), I think this conclusion cannot be drawn from this study. These proteins may rather be describing protein level mechanisms occurring during or more likely following infarct, and may differ in their nature to protein abnormalities which may be present prior to stroke. This needs to be clear in the discussion. As it stands this is a major conclusion of the study which I think is misleading. This is important as future research may (as the authors elude to) concentrate on developing biomarker tests (including those to be used as point of care tests) to understand risk of stroke sequelae and inform clinical management.

2. Minor: I agree with the authors that more detailed analysis of the nature of stroke would contribute to the paper and therefore its absence is a limitation of the study. In the least would it be possible for more detailed information on the time of onset of the stroke (either from clinical or radiological findings) in order to ensure the findings here differentiated stroke which occurred as part of the TBM presentation, and those which may have occurred as part of a separate illness? It states in the manuscript that those with radiological evidence of infarct were assigned to the TBM with stroke group, however there is no indication that more rigorous analysis of the clinical/radiological data included within this group were only those with stroke occurring as part of this episode of TBM.

3. Minor: Baseline demographics are relatively sparse. Were there other noticeable differences between the TBM-stroke and TBM-no stroke group in terms of clinical presentation ie proportion of definite vs probable cases, severity of presentation/BMRC grade, other radiological features (eg hydrocephalus, tuberculomas etc)?

Reviewer #2: This is a pilot study examining potential biomarkers of stroke in patients with tuberculous meningitis (TBM). Stroke is a key factor contributing to poor outcomes in TBM yet is currently under-studied and poorly understood. Therefore, this is a worthwhile study in starting to address some of the unanswered questions about TBM-associated stroke, including the need for diagnostic and predictive tools that can guide patient management. The pilot data generated are certainly interesting and could serve as valuable preliminary data to inform future studies that aim to address the question of biomarkers on a larger scale. Some of the key limitations to the study have been addressed, including the small sample size, absence of infarct classification, and lack of serial sampling. However, I think there are other important limitations that also require attention, including the combination of CT and MRI images and the lack of serial imaging to examine infarct evolution. These factors could significantly change the stroke and non-stroke groups and can therefore not be overlooked. I have also raised some questions about the choice of control group and the lack of a reported association between the controls and cases (unless I missed this??).

Methods

• What was the definition of stroke on imaging? Ie: were small lacunar infarcts considered equally with large vascular territory infarcts? Similarly, were only established infarcts considered, or also evolving/acute infracts as would be seen on DWI?

• The control group is quite heterogenous in their pathology and it seems like some had neurological disease while others did not – were CSF samples collected from all these controls? What was the eligibility criteria for the control group?

Results

• On page 15, line 180 the authors refer to the AUC of 24 of the 69 markers, which 24 markers are these and how were they selected?

• What was the difference in biomarker concentrations between the TBM and control cohorts in CSF and blood? A possible caution for the analysis would be the heterogeneity of the control group with some having CNS pathology while others do not – this may factor into the selection of controls for comparison or the interpretation of findings…

• Also, from Figure 2 (and 3) it looks like the biomarker concentrations between the TBM with stroke and not-TBM cohorts are very similar – was a statistical comparison done between these groups? If so, what were the results? I think this is an important comparison to make so that the specificity of these biomarkers for TBM stroke can be established.

Discussion

• The authors acknowledge the key limitations of this study, ie: small sample size, single time point testing and the grouping of heterogenous imaging data into a homogenous group. Further limitations the authors should address include 1) that they used a combination of CT and MRI when we know from previous work done by this group that MRI has better sensitivity in showing location, number and temporal resolution of infarcts (Pienaar et al, Childs Nervous System, 2004) , 2) that they did not look at the evolution of infarcts over time – TBM patients can show signs of infarction over the first weeks of treatment that are not present on admission (Rohlwink et al, Pediatric Infectious Disease Journal, 2017), this would have been a key analysis to establishing the predictive power of their biomarker signatures for stroke

• I found it interesting that the multi-marker signatures with high predictive value in CSF and blood largely comprise biomarkers that did not come up as significant on the stroke vs non-stroke comparison; why do the authors think this may be the case?

Tables and figures

Table 2 and 3

• These tables are a bit too full, I would suggest editing the column headings to make them shorter

6. PLOS authors have the option to publish the peer review history of their article (what does this mean?). If published, this will include your full peer review and any attached files.

Reviewer #1: No

Reviewer #2: No

---

## [Author Response · Author response to Decision Letter 0]

14 Apr 2021

Point by Point Responses to reviewers

Journal Requirements:

Responses: We have made changes to the reference list to include a reference suggested by reviewer 1. The following reference was included:

[34] Pienaar M, Andronikou S, van Toorn R. MRI to demonstrate diagnostic features and complications of TBM not seen with CT. Childs Nerv Syst. 2009;25: 941–947. doi:10.1007/s00381-008-0785-3

Response: We have ensured that our manuscript meets PLOS ONE’s style requirements, including for file naming.

2. In your Methods section, please provide additional information about the participant recruitment method and the demographic details of your participants. Please ensure you have provided sufficient details to replicate the analyses such as:

a) a description of any inclusion/exclusion criteria that were applied to participant selection,

Response: The study setting section under methods of the manuscript was revised to clearly state the inclusion/exclusion criteria. Lines 86-92.

b) a statement as to whether your sample can be considered representative of a larger population, and

Response: We have indicated in the methods section that our sample is a representative of the typical patients from our study community (Lines 85-86)

c) a brief description of how participants were recruited in the original study.

Response: We have revised the study setting section to describe how the participants were recruited in the original study lines 81-87 “Briefly, in these studies the participants were enrolled at Tygerberg Academic Hospital, Cape Town, South Africa between November 2016 and November 2017. Children with suspected TBM are referred from primary care day hospitals, district and secondary level hospitals to our institution to establish the diagnosis of TBM and to treat the complications associated with advanced disease (stage 2 and 3 TBM, e.g. hydrocephalus). We enrolled 47 children presenting with signs and symptoms suggestive of meningitis and requiring routine diagnostic assessment including lumbar puncture for CSF investigations”

Response: We have uploaded the minimal anonymized data set necessary to replicate our study findings as a supporting information file during re-submission (S1 File)

4.We note that you have a patent relating to material pertinent to this article. Please provide an amended statement of Competing Interests to declare this patent (with details including name and number), along with any other relevant declarations relating to employment, consultancy, patents, products in development or modified products etc. Please confirm that this does not alter your adherence to all PLOS ONE policies on sharing data and materials, as detailed online in our guide for authors http://journals.plos.org/plosone/s/competing-interests by including the following statement: "This does not alter our adherence to PLOS ONE policies on sharing data and materials.” If there are restrictions on sharing of data and/or materials, please state these. Please note that we cannot proceed with consideration of your article until this information has been declared.

Response: We have amended statement of Competing Interests to declare the patent and confirmed that this does not alter our adherence to all PLOS ONE policies on sharing data and materials. Our amended statement is as follows: NC, CM, GW and RS are listed as inventors on an International Patent Application entitled “Cerebrospinal fluid and blood-based biomarkers for the diagnosis of tuberculosis meningitis” (PCT/IB2019/054259), filing date: 23 May 2019. NC and GW are listed as inventors on another patent application entitled “Method for diagnosing tuberculous meningitis” (PCT/IB2015/052751), filing date: 15 April 2015. These applications do not generate any royalties for the inventors. These does not alter our adherence to PLOS ONE policies on sharing data and materials. 

5.We noticed you have some minor occurrence of overlapping text with the following previous publication(s), which needs to be addressed:

https://scholar.sun.ac.za/handle/10019.1/106647

In your revision ensure you cite all your sources (including your own works), and quote or rephrase any duplicated text outside the methods section. Further consideration is dependent on these concerns being addressed.

Response: We have rephrased the duplicated texts outside the methods section and have referenced the original publication that appears in https://scholar.sun.ac.za/handle/10019.1/106647 on the sections that were adapted from this publication.

Original reference: 

17. Manyelo CM, Solomons RS, Snyders CI, Manngo PM, Mutavhatsindi H, Kriel B, et al. Application of Cerebrospinal Fluid Host Protein Biosignatures in the Diagnosis of Tuberculous Meningitis in Children from a High Burden Setting. Giovarelli M, editor. Mediators of Inflammation. 2019;2019: 7582948. doi:10.1155/2019/7582948

Reviewer's Responses to Questions

Comments to the Author

1. Review Comments to the Author

Reviewer #1: Many thanks for asking me to review this paper. The manuscript describes analysis undertaken following your larger studies which have investigated a wider spectrum of protein changes in patients with TBM and those without. I agree with the authors that analysis of proteins related to stroke, an often disabling or life threatening sequelae in TBM, warranted further analysis. Knowledge gained here will contribute to our understanding of the mechanisms which underpin stroke pathogenesis within TBM. However, I have a few comments to be addressed:

1. Major: One of the conclusions of the proteins identified have the potential to contribute to management by 'predicting' stroke and therefore identifying those who may benefit from preventative therapy such as anti-platelet therapy. Given the design of this study where proteins were identified in blood and CSF samples of patients at baseline who either had or had not already developed stroke (ie imaging was performed at the same timepoint as blood/CSF samples taken), I think this conclusion cannot be drawn from this study. These proteins may rather be describing protein level mechanisms occurring during or more likely following infarct, and may differ in their nature to protein abnormalities which may be present prior to stroke. This needs to be clear in the discussion. As it stands this is a major conclusion of the study which I think is misleading. This is important as future research may (as the authors elude to) concentrate on developing biomarker tests (including those to be used as point of care tests) to understand risk of stroke sequelae and inform clinical management.

Response: We thank the reviewer for this comment and the suggestions. Indeed the proteins identified in this study may describe protein level mechanisms occurring during or more likely following already established infarcts and may differ from the levels that may be seen prior to stroke. The major conclusion of our initially submitted version may be misleading, and we have considered the reviewer’s suggestion and revised our conclusion. The protein biomarkers identified in this study may be useful for detection or indication of stroke in patients diagnosed with TBM at admission. This may especially be beneficial in settings where neuroimaging is not available. The future research could then concentrate on evaluating the abilities of these proteins and other proteins for prediction of future stroke, by following TBM patients over time, and observe if those predicted to have stroke will develop stroke over a period of time.

2. Minor: I agree with the authors that more detailed analysis of the nature of stroke would contribute to the paper and therefore its absence is a limitation of the study. In the least would it be possible for more detailed information on the time of onset of the stroke (either from clinical or radiological findings) in order to ensure the findings here differentiated stroke which occurred as part of the TBM presentation, and those which may have occurred as part of a separate illness? It states in the manuscript that those with radiological evidence of infarct were assigned to the TBM with stroke group, however there is no indication that more rigorous analysis of the clinical/radiological data included within this group were only those with stroke occurring as part of this episode of TBM.

Response: In the 14 children with TBM and stroke, 3 children presented 2-14 days prior to admission which was considered compatible with the prolonged symptom duration seen in TBM, and in 6 children hemiplegia was the reason for admission. In the 5 children without hemiplegia, acute radiological infarction was detected on admission CT brain (within 24-48 hours).

3. Minor: Baseline demographics are relatively sparse. Were there other noticeable differences between the TBM-stroke and TBM-no stroke group in terms of clinical presentation ie proportion of definite vs probable cases, severity of presentation/BMRC grade, other radiological features (eg hydrocephalus, tuberculomas etc)?

Response: We thank the reviewer for this comment. We have revised the baseline demographics in Table 1 and have now included other patient characteristics/features including as ‘Admission characteristics’ and ‘other radiological features’.

Reviewer #2: This is a pilot study examining potential biomarkers of stroke in patients 1with tuberculous meningitis (TBM). Stroke is a key factor contributing to poor outcomes in TBM yet is currently under-studied and poorly understood. Therefore, this is a worthwhile study in starting to address some of the unanswered questions about TBM-associated stroke, including the need for diagnostic and predictive tools that can guide patient management. The pilot data generated are certainly interesting and could serve as valuable preliminary data to inform future studies that aim to address the question of biomarkers on a larger scale. Some of the key limitations to the study have been addressed, including the small sample size, absence of infarct classification, and lack of serial sampling. However, I think there are other important limitations that also require attention, including the combination of CT and MRI images and the lack of serial imaging to examine infarct evolution. These factors could significantly change the stroke and non-stroke groups and can therefore not be overlooked. I have also raised some questions about the choice of control group and the lack of a reported association between the controls and cases (unless I missed this??).

Response: We thank the reviewer for reviewing our work and for all the suggestions. Indeed our work is a pilot study examining the differences in biomarker concentration between TBM patients with stroke and those without stroke, and to further look at potential biomarkers of stroke in patients with TBM. We have considered all the comments and suggestions raised by the reviewer and have responded to all the points below.

Methods

• What was the definition of stroke on imaging? Ie: were small lacunar infarcts considered equally with large vascular territory infarcts? Similarly, were only established infarcts considered, or also evolving/acute infracts as would be seen on DWI?

Response: Radiological arterial ischemic infarction was defined as neuroimaging evidence of infarction, i.e. interruption of blood flow eventually resulting in focal encephalomalacia. Mostly small areas of arterial ischemic infarction in the territory of the middle cerebral artery perforators i.e basal ganglia and internal capsule, were observed. When CT was performed established arterial ischemic infarcts were considered, and when MRI was performed both established and evolving arterial ischemic infarction were considered (Line 103-107 in the revised version)

• The control group is quite heterogenous in their pathology and it seems like some had neurological disease while others did not – were CSF samples collected from all these controls? What was the eligibility criteria for the control group?

Response: All the study participants were eligible for inclusion into this study if they presented with signs and symptoms suggestive of meningitis, and requiring assessment to establish TBM diagnosis or an alternative diagnosis. The CSF samples were collected for the purpose of routine diagnostic assessment, and additional CSF samples were collected from each patient for the purpose of this study. So, the CSF collection (lumbar puncture) was not done specifically for this study. Written informed consent for inclusion in this study was obtained from the caregivers.

Results

• On page 15, line 180 the authors refer to the AUC of 24 of the 69 markers, which 24 markers are these and how were they selected?

Response: We thank the reviewer for this comment. The 24 of the 69 markers referred to here are listed in S2 Table in the initial submission. To make this section more clear, the 24 of the 69 markers were listed in the revised manuscript (Lines 193-197). The 24 markers were selected on the basis of area under the curve, whereby the markers were arranged (sorted) according to the highest AUC, and those with AUC of at least 0.70 were considered. As similar issue would apply to line 236 (of previous version), we also corrected this section by listing the 23 of the 69 serum markers on the revised manuscript (Line 251-255).

• What was the difference in biomarker concentrations between the TBM and control cohorts in CSF and blood? A possible caution for the analysis would be the heterogeneity of the control group with some having CNS pathology while others do not – this may factor into the selection of controls for comparison or the interpretation of findings…

Response: The difference in biomarker concentrations between the TBM and control cohorts in CSF and blood were not reported in the current manuscript. We previously reported on the differences in biomarker concentrations between TBM and control cohorts from the same study participants in CSF (Reference: Manyelo et al., Mediators of Inflammation. 2019) and blood (Reference: Manyelo et al., Front Pediatr. 2019) In the current manuscript we focused specifically on the differences in biomarker concentrations between TBM patients with stroke and TBM patients without stroke, and further included the no-TBM controls, which were all part of the previous cohort.

We agree with the reviewer that the heterogeneity of the control group may be a possible caution, however we were including children who were presenting with signs and symptoms suggestive of TBM, as in a practical clinical setting all these children may be assessed to establish a TBM diagnosis or to rule out TBM. We then measured biomarker concentrations in all the study participants prior to final diagnosis of TBM. Upon final diagnosis, all the children with alternative diagnosis were classified as no-TBM. It may be good in future to include control cohort with CNS pathology such as viral meningitis, bacterial meningitis, and fungal meningitis.

• Also, from Figure 2 (and 3) it looks like the biomarker concentrations between the TBM with stroke and not-TBM cohorts are very similar – was a statistical comparison done between these groups? If so, what were the results? I think this is an important comparison to make so that the specificity of these biomarkers for TBM stroke can be established.

Response: Thank you for this comment. The statistical comparison for TBM with stroke and not-TBM were done, however as the main aim was to compare the differences in biomarker concentration between TBM patients with stroke and without stroke, we did not report the statistical comparison for TBM with stroke and not-TBM in the submitted version. However, we have considered the reviewers suggestion and have now included the statistical comparison of CSF and serum biomarker concentrations between TBM with stroke and not-TBM in S2 and S3 tables (Revised S2 and S3 Tables were resubmitted), in the revised version. Furthermore, texts were inserted in lines 190-191, and lines 249-250 to mention how many CSF and serum proteins were statistically different between TBM-related stroke and not-TBM, respectively. As the aim was to mainly identify biomarkers that are different between TBM with stroke compared to TBM without stroke, and can be used to detect stroke or guide therapy in patients who are finally diagnosed with TBM, we then put focus of our results on TBM with stroke compared to TBM without stroke. Thus, the specificity of these biomarkers may only be important between TBM patients with stroke and TBM without stroke.

Discussion

• The authors acknowledge the key limitations of this study, ie: small sample size, single time point testing and the grouping of heterogenous imaging data into a homogenous group. Further limitations the authors should address include 1) that they used a combination of CT and MRI when we know from previous work done by this group that MRI has better sensitivity in showing location, number and temporal resolution of infarcts (Pienaar et al, Childs Nervous System, 2004) , 2) that they did not look at the evolution of infarcts over time – TBM patients can show signs of infarction over the first weeks of treatment that are not present on admission (Rohlwink et al, Pediatric Infectious Disease Journal, 2017), this would have been a key analysis to establishing the predictive power of their biomarker signatures for stroke

Responses:

1) The authors acknowledge that the lack of MRI imaging, which provides much more detailed neuroimaging than CT, is a limitation. However, in TB endemic settings, the cost of MRI is often prohibitive. We definitely aim, with careful planning, to use MRI as the primary imaging modality in prospective studies. (Lines 347-351 in the revised version)

2) We thank the reviewer for this comment, and we have revised the discussion to include this as one of the limitations of our study (Line 351-355). We acknowledge that it would be good in the future study to follow-up patients and look at the evolution of infarcts over time. As the reviewer has suggested, this will allow to establish the predictive power of the biomarker signatures over time. The patient were recruited into this study as part of our previous studies in which we collected CSF and blood samples, and neuroimaging data only at baseline. This work will guide future work in which we could follow-up patients and look at evolution of infarcts and biomarker signature predictive abilities over time. 

• I found it interesting that the multi-marker signatures with high predictive value in CSF and blood largely comprise biomarkers that did not come up as significant on the stroke vs non-stroke comparison; why do the authors think this may be the case?

Response: We thank the reviewer for raising this and we have acknowledged this as a possible limitation in the revised version (line 358-363). Indeed the multi-marker signatures comprised largely of biomarkers that were not significantly different between TBM-related stroke and no stroke. This may be due to the smaller sample size, and hence it will be good to further investigate this biosignatures in a larger cohort, to determine if accuracies of the multi-marker signature were true results. However, we put more focus in the individual biomarkers that were significantly different between stroke and no-stroke and can contribute to the understanding of biology of stroke, as well as indication of stroke at baseline. However, all our findings require further research in a larger study.

Tables and figures

Table 2 and 3

• These tables are a bit too full, I would suggest editing the column headings to make them shorter

Response: We thank the reviewer for this suggestion. We have edited the column headings on the tables to make them shorter.

---

## [Editor Report · Decision Letter 1]

19 Apr 2021

Serum and cerebrospinal fluid host proteins indicate stroke in children with tuberculous meningitis

PONE-D-21-04815R1

Dear Dr. Solomons,

We’re pleased to inform you that your manuscript has been judged scientifically suitable for publication and will be formally accepted for publication once it meets all outstanding technical requirements.

Kind regards,

Katalin Andrea Wilkinson, PhD

Academic Editor

PLOS ONE
---

## [Editor Report · Acceptance letter]

22 Apr 2021

PONE-D-21-04815R1 

Serum and cerebrospinal fluid host proteins indicate stroke in children with tuberculous meningitis 

Dear Dr. Solomons:

I'm pleased to inform you that your manuscript has been deemed suitable for publication in PLOS ONE. Congratulations! Your manuscript is now with our production department. 

Kind regards, 

on behalf of

Associate Professor Katalin Andrea Wilkinson 

Academic Editor

PLOS ONE